# Stochastic Layer-Wise Shuffle for Improving Vision Mamba Training

**Zizheng Huang** [1][2]  **Haoxing Chen** [3]  **Jiaqi Li** [4]  **Jun Lan** [3]  **Huijia Zhu** [3]  **Weiqiang Wang** [3]  **Limin Wang** [1][5]

## Abstract

Recent Vision Mamba (Vim) models exhibit nearly linear complexity in sequence length, making them highly attractive for processing visual data. However, the training methodologies and their potential are still not sufficiently explored. In this paper, we investigate strategies for Vim and propose Stochastic Layer-Wise Shuffle (SLWS), a novel regularization method that can effectively improve the Vim training. Without architectural modifications, this approach enables the non-hierarchical Vim to get leading performance on ImageNet-1K compared with the similar type counterparts. Our method operates through four simple steps per layer: probability allocation to assign layer-dependent shuffle rates, operation sampling via Bernoulli trials, sequence shuffling of input tokens, and order restoration of outputs. SLWS distinguishes itself through three principles: *(1) Plug-and-play:* No architectural modifications are needed, and it is deactivated during inference. *(2) Simple but effective:* The four-step process introduces only random permutations and negligible overhead. *(3) Intuitive design:* Shuffling probabilities grow linearly with layer depth, aligning with the hierarchical semantic abstraction in vision models. Our work underscores the importance of tailored training strategies for Vim models and provides a helpful way to explore their scalability. Code and models are available at the open source URL.

## 1. Introduction

Vision Transformers (ViTs) (Dosovitskiy et al., 2021; Liu et al., 2021; Dong et al., 2022; He et al., 2022; Bao et al., 2022) have achieved remarkable performance in modeling visual data, yet their quadratic complexity w.r.t. sequence length (Katharopoulos et al., 2020) remains a significant drawback. In contrast, recent advances in State Space Models (SSMs) (Kalman, 1960; Gu et al., 2021a;b; Smith et al., 2023) offer potentially more efficient sequence-based vision encoders (Zhu et al., 2024; Smith et al., 2023; Liang et al., 2024; Zhang et al., 2024b; Li et al., 2024b). Among these, Mamba (Gu & Dao, 2023; Dao & Gu, 2024) stands out for its hardware-friendly design and selective scan computation, enabling near-linear complexity for longer sequences and prompting adoption in various vision tasks (Zhu et al., 2024; Liu et al., 2024c; Wang et al., 2024; Yang et al., 2024a). Extensions that incorporate 2-D scanning paths and visual priors (Zhu et al., 2024; Li et al., 2024a; Huang et al., 2024; Zhang et al., 2024a; Li et al., 2025; Tang et al., 2024) have demonstrated competitive or even superior performance compared to ViTs (Liang et al., 2024; Wu et al., 2024; Yue & Li, 2024). Such improvements, observed across supervised pre-training and diverse downstream applications (Chen et al., 2024; Patro & Agneeswaran, 2024; Phung et al., 2024), highlight Mamba's potential as an efficient, scalable foundation for visual processing (Yang et al., 2024b; Liu et al., 2024a; Xiao et al., 2024).

Initial efforts to scale up Vision Mamba (Vim) models were hindered by overfitting issues (Zhu et al., 2024; Ren et al., 2024; Li et al., 2025), causing performance degradation and even model collapse. In addition, the non-hierarchical Vim architecture further complicates the pursuit of higher accuracy (Li et al., 2024a; Tang et al., 2024). Although a limited number of supervised and unsupervised strategies (Wang et al., 2024; Liu & Yi, 2024) have successfully trained and scaled certain Mamba-based models to huge sizes (Ren et al., 2024), recent research has moved beyond mere model enlargement toward broader, more robust improvements. Nevertheless, more efficient training methodologies are still urgently needed to overcome challenges like overfitting and narrow the performance gap with leading architectures such as ViT on ImageNet-1k (He et al., 2022; Wei et al., 2022a; Hou et al., 2022; Peng et al., 2022), where MambaMLP-L (Ren et al., 2024) (84.5%) still trails MAE-L (He et al., 2022) (85.9%).

In this paper, we focus on training methods for Vim models and propose a *Stochastic Layer-Wise Shuffle* regularization algorithm that effectively mitigates overfitting and boosts

---

[1]State Key Lab of Novel Software Technology, Nanjing University [2]Shanghai Innovation Institute [3]Independent Researcher [4]China Mobile Research Institute [5]Shanghai AI Lab. Correspondence to: Limin Wang <lmwang@nju.edu.cn>.

performance in large-scale Vim architectures. Concretely, the algorithm unfolds in four steps at each layer's forward pass: (1) probability allocation to assign layer-dependent shuffle rates, (2) operation sampling via a Bernoulli trial, (3) shuffling the input token sequence, and (4) restoring the output sequence to its original order. The underlying rationale is that deeper layers, which need higher-level semantic representations, can tolerate greater perturbations in token positions, whereas shallower layers should remain sensitive to low-level information. Restoring the sequence order prevents recursive effects for later layers. We verify the effectiveness of this method in both supervised classification settings and a pre-training plus fine-tuning paradigm. The main contributions of this paper are summarized as follows:

(1) We present a *Stochastic Layer-Wise Shuffle* regularization algorithm for non-hierarchical Vision Mamba models. This plug-and-play method effectively mitigates overfitting, introduces minimal overhead, and requires no changes to the underlying architecture.

(2) In a supervised setting, we show that the algorithm successfully addresses overfitting in large-scale models, boosting performance in visual classification and downstream dense prediction tasks (e.g., ADE20K segmentation and COCO detection).

(3) We further integrate masked feature distillation into the Vim pretraining process, demonstrating Vision Mamba can also beneficial from a semantic-rich frozen tokenizer. Notably, incorporating SLWS achieves 87.6% accuracy on ImageNet-1K, establishing a new state-of-the-art for Vision Mamba models on this benchmark.

## 2. Related Work

**Vision Backbones**  In the field of computer vision, the exploration of efficient and scalable backbone architectures has led to significant advancements (He et al., 2016; Krizhevsky et al., 2017; Dosovitskiy et al., 2021; Zhu et al., 2024), primarily driven by CNNs (Simonyan & Zisserman, 2015; Li et al., 2019; Liu et al., 2022b) and ViTs (Dosovitskiy et al., 2021; Liu et al., 2021; Wang et al., 2021) recently. Initially, CNNs serve as the foundation and have evolved into deeper architectures, such as AlexNet (Krizhevsky et al., 2017), VGG (Simonyan & Zisserman, 2015), and ResNet (He et al., 2016). Various studies have introduced advanced operators, architectures, and attention mechanisms to improve the effectiveness of models such as SENet (Hu et al., 2018) and SKNet (Li et al., 2019). The continuous refinement of convolutional layers has resulted in architectures like RepLKNet (Ding et al., 2022) and ConvNeXt (Liu et al., 2022b), which offer improved scalability and accuracy. Despite significant advancements, CNNs primar-

ily focus on exploiting spatial locality, making assumptions about feature locality, translation, and scale invariance.

The introduction of ViT (Dosovitskiy et al., 2021) marks a turning point. Adapted from the NLP community (Vaswani et al., 2017), ViTs treat images as sequences of flattened 2D patches to capture global relationships (Liu et al., 2022a; Wang et al., 2021). As ViTs evolved, models like DeiT addressed optimization challenges (Touvron et al., 2021; He et al., 2022), while others introduced hierarchical structures and convolution operations to incorporate inductive biases of visual perception (Liu et al., 2021; Wang et al., 2021; 2022). These modifications allow for better performance across diverse visual tasks, although at the cost of added complexity in the models. Recently, there has been a trend of reverting to the original, plain ViT architecture due to its simplicity and flexibility in pre-training and fine-tuning across tasks (Bao et al., 2022; Xia et al., 2022; Carion et al., 2020; Cheng et al., 2022). However, one of the major challenges is the quadratic complexity of the self-attention mechanism (Katharopoulos et al., 2020; Zhu et al., 2023) limits the number of visual tokens that can be processed thereby impacting efficiency.

**State Space Vision Models**  Early state space transformations (Gu et al., 2021a;b; Smith et al., 2023; Gu et al., 2023), inspired by continuous state models and bolstered by HiPPO initialization (Gu et al., 2020), showcased the potential for handling extensive dependency problems (Nguyen et al., 2023; Tallec & Ollivier, 2018). To overcome computational and memory issues, S4 (Gu et al., 2021a) enforced diagonal structure on the state matrix, while S5 (Smith et al., 2023) introduced parallel scanning to enhance efficiency further. The Mamba model (Gu & Dao, 2023; Dao & Gu, 2024) stands out for its novel approach to SSMs. By parameterizing the state space matrices as projections of input data, Mamba proposes the more flexible selective scanning.

While ViTs and CNNs have laid a robust foundation for various visual tasks, Mamba offers a unique potential due to the ability to scale linearly with sequence length (Patro & Agneeswaran, 2024; Zhu et al., 2024; Nguyen et al., 2022; Lieber et al., 2024). S4ND (Nguyen et al., 2022) is the pioneering effort to integrate SSM into visual applications. However, the straightforward expansion did not efficiently capture image information. This gap led to further innovations in hybrid CNN-SSM hierarchical architecture, such as U-Mamba (Liu et al., 2024b), VMamba (Liu et al., 2024c) and MambaMixer(Behrouz et al., 2024). Recent efforts have sought to build generic vision backbones purely based on SSMs without relying on attention mechanisms (Zhu et al., 2024; Li et al., 2024a; Wang et al., 2024; Ren et al., 2024). Vision Mamba model, built by sequentially stacking Mamba blocks, has been shown to outperform ViT or perform on par in small model sizes (Wang et al., 2024; Liu & Yi, 2024;

Yang et al., 2024a). There are also some work exploring to refine the scanning method in Vim for visual data (Yang et al., 2024a; Li et al., 2024a; Huang et al., 2024; Chen et al., 2024; Tang et al., 2024; Pei et al., 2024). Nevertheless, Vims are stuck into issues like overfitting and have a noticeable performance gap compared to ViT in large sizes.

**Training Methodologies** To improve the training and generalization of deep models, various regularization techniques have been developed over the past years. Normalizations (Ioffe & Szegedy, 2015; Ulyanov et al., 2016; Wu & He, 2018) are proven to be effective for speeding up the convergence, in which the Layer Normalization (Ba et al., 2016) and RMSNorm (Zhang & Sennrich, 2019) are popular in training of large models. The family of data augmentations (Cubuk et al., 2020; Hoffer et al., 2020; Yun et al., 2019; Zhang et al., 2018) helps to produce more robust representations and enhance performance. Stochastic depth and drop path (Huang et al., 2016; Larsson et al., 2016) drop the connection in the block level, which can not only overcome overfitting but also decrease the training cost. Weight decay (Krogh & Hertz, 1991; Loshchilov & Hutter, 2019) is commonly adopted for mitigating overfitting as well in a weight-penalizing manner. Besides, the earlier Dropout approach (Srivastava et al., 2014) introduces disturbance by dropping hidden units. They have played roles in various network training scenarios.

*When it comes to vision models*, numerous training strategies have been proposed beyond supervised classification. Early self-supervised methods relied on surrogate tasks such as jigsaw puzzles (Noroozi & Favaro, 2016) predicting spatial context (Doersch et al., 2015), while subsequent contrastive approaches like SimCLR (Chen et al., 2020a), MoCo (He et al., 2020; Chen et al., 2020c; 2021)), and iBoT(Zhou et al., 2022) effectively trained both CNNs and ViTs by leveraging instance discrimination. More recently, masked pre-training techniques begin from MAE (He et al., 2022; Tong et al., 2022) and BEiT (Bao et al., 2022) have shown remarkable potential for scaling ViT models. These kinds of methods reconstruct raw pixels or discrete tokens to learn semantic-rich embeddings (Xie et al., 2022; Chen et al., 2020b). Additionally, with a Self-EMA or frozen tokenizer, masked feature distillation methods (Peng et al., 2022; Hou et al., 2022; Fang et al., 2023; Baevski et al., 2022) can further elevate their generalization and performance of ViTs. In this strategy, the student model processes remaining patches after masking and is trained with the teacher target, which showcases superior efficiency and performance (Fang et al., 2023; Li et al., 2023; Peng et al., 2023).

*For non-hierarchical Vim models*, several training methods extend beyond scanning-based approaches. Vim-F (Zhang et al., 2024c) explores frequency-domain training to enhance the global receptive field, showing improvements for Tiny and Small Vim models. Mamba-Reg (Wang et al., 2024) introduces "registers" (a group of extra [CLS] tokens) to mitigate high-norm outliers, enabling Mamba-Reg to outperform ViTs under supervised classification. Meanwhile, ARM (Ren et al., 2024) and MAP (Liu & Yi, 2024) adopt autoregressive pipelines to further scale up Vim models. Despite these advances, a noticeable performance gap remains between Vim and ViT, highlighting the urgent need for continued exploration of Vim's capabilities.

## 3. Methodology

In this section, we propose Stochastic Layer-Wise Shuffle Regularization (SLWS) for supervised training of non-hierarchical Vim models, along with a brief introduction of masked distillation strategy employed for pre-training. We first present the preliminaries in the following subsections to establish foundational concepts.

### 3.1. Preliminaries

**State Space Model** (SSM) (Gu et al., 2021a;b) is originally designed for modeling continuous-time systems by projecting 1-D input stimulation $x(t)$ to the output signal $y(t)$ via hidden state $h(t) \in \mathbb{R}^n$. Formally, SSM is expressed with the subsequent ordinary differential equation (ODE) as follows:

$$
\begin{aligned}
h'(t) &= \mathbf{A}h(t) + \mathbf{B}x(t), \\
y(t) &= \mathbf{C}h(t) + \mathbf{D}x(t),
\end{aligned}
\tag{1}
$$

where $\mathbf{A} \in \mathbb{R}^{n \times n}$ denotes the system's evolutionary matrix, with $\mathbf{B} \in \mathbb{R}^{n \times 1}$, $\mathbf{C} \in \mathbb{R}^{1 \times n}$ and $D$ are projection parameters. In a discrete system scenario, the above SSM is discretized by a timescale parameter $\mathbf{\Delta}$, transforming the expressions of $\mathbf{A}$ and $\mathbf{B}$ into their discrete equivalents $\bar{\mathbf{A}}$ and $\bar{\mathbf{B}}$. In Mamba models, such conversion is implemented with the Zero-Order Hold (ZOH) rule, which is expressed as follows:

$$
\begin{aligned}
\bar{\mathbf{A}} &= \exp(\mathbf{\Delta}A), \\
\bar{\mathbf{B}} &= \mathbf{\Delta}A^{-1}(\exp(\mathbf{\Delta}A - \mathbf{I})) \cdot \mathbf{\Delta}B.
\end{aligned}
\tag{2}
$$

Then, a sequential input $\{x_i\}_{i=1}^{L}$ is mapped via this discretized system to its output $\{y_i\}$ as:

$$
\begin{aligned}
h'_i &= \bar{\mathbf{A}}h_{i-1} + \bar{\mathbf{B}}x_i, \\
y_i &= \mathbf{C}h'_i + \mathbf{D}x_i.
\end{aligned}
\tag{3}
$$

Mamba (Gu & Dao, 2023) designs the $\mathbf{B}$, $\mathbf{C}$, and $\mathbf{\Delta}$ to be input-dependent to improve the intrinsic capacity for contextual sensitivity and adaptive weight modulation. Besides, a Selective Scan Mechanism is ensembled in for efficient computation. To this end, for a Vim (Zhu et al., 2024) block (or layer) $s_\ell$, it includes an SSM branch, whose output is

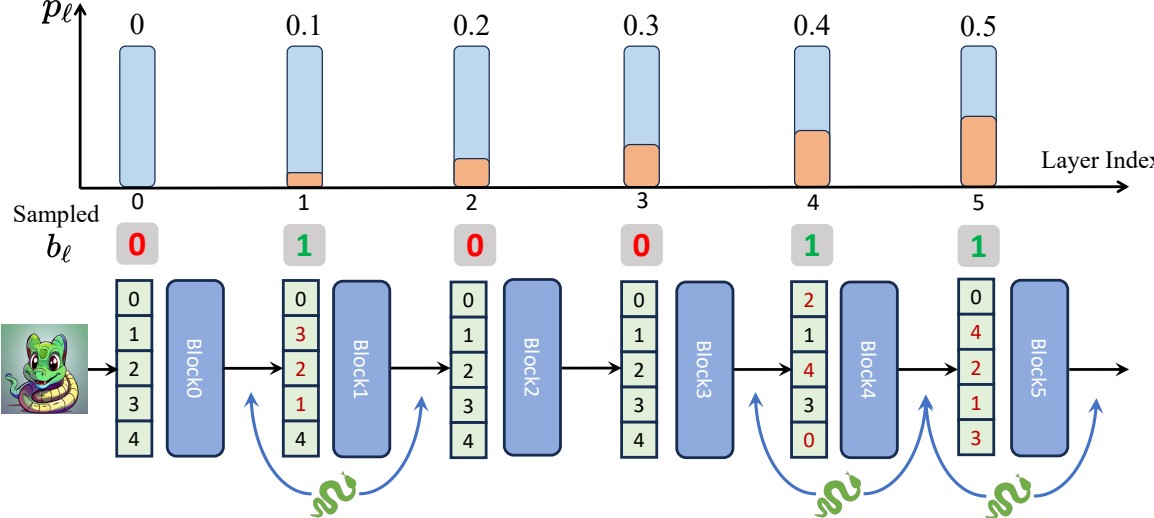

Figure 1: **Stochastic Layer-Wise Shuffle Regularization (SLWS).** Deeper layers are assigned larger probabilities for shuffle regularization to enhance positional transformation invariance. The variable $b_\ell$ is sampled based on these probabilities to determine whether to execute regularization. SLWS only involves sequence permutation and restoration, and is not applied during inference. The snake icon indicates where regularization is performed.

multiplied by the result of another gated branch to produce the final output sequence $X_\ell \in \mathbb{R}^{T \times D}$. Thus, the corresponding forward process of non-hierarchical Vim (without downsampling) is expressed in the following form:

$$X_\ell = s_\ell \left( X_{\ell-1} \right). \tag{4}$$

**Masked Feature Distillation** (MFD) techniques enhance pre-training by masking a significant portion of image patches and subsequently reconstructing the targets using the unmasked regions as input. Methods such as MAE (He et al., 2022) have been proven effective in training foundational Vision Transformers (ViTs) without relying on labeled data. Further research has shown that employing feature-level targets can lead to additional improvements, including the use of HOG features (Wei et al., 2022a), Self-EMA (Baevski et al., 2022), CLIP embeddings (Radford et al., 2021; Wei et al., 2022b; Hou et al., 2022), and discrete tokens (Peng et al., 2022). The MFD process can be formulated as follows:

$$\min_{X} \mathbb{E} \text{ dist} \left[ \mathcal{T}(X), d \left( f \left( X^v \right) \right) \right], \tag{5}$$

where $\mathcal{T}$ represents the teacher tokenizer, $f$ denotes the student model, and $X^v$ refers to the remained visible parts. dist $[\cdot, \cdot]$ is the selected distance function.

### 3.2. Stochastic Layer-Wise Shuffle

As formulated above, the SSM-based Mamba was originally proposed for sequence modeling but does not naturally adapt to two-dimensional image data, where patch

sequences are non-causal. Several previous studies have integrated different scanning strategies into Mamba layers to better capture spatial context (Zhu et al., 2024; Liu et al., 2024c; Yang et al., 2024a; Li et al., 2024a; Tang et al., 2024). Nevertheless, these methods remain reliant on simple 1-D corner-to-corner scanning and often suffer from overfitting. To address these limitations, we propose Stochastic Layer-Wise Shuffle (SLWS), a regularization technique guided by the following insights:

(1) Fixed corner-to-corner sequential or regional scanning in Vim does not naturally align with the need to capture both local and global spatial correlations.

(2) Deeper layers of a vision encoder should learn higher-level semantic representations, while shallower layers primarily encode low-level information.

(3) Achieving stronger semantic perception in deeper layers requires *transformation invariance* for patch positions, whereas shallower layers must preserve *positional sensitivity*.

(4) Introducing stochastic perturbations into the sequential structure can increase *task complexity*, potentially mitigating overfitting, but also contributes to *simulate diverse spatial contexts*.

(5) Besides designing layer-dependent for differing semantic requirements, a sequence restoration step ensures that subsequent layers receive inputs in the original order, thus avoiding unnecessary disruptions.

**Random Shuffle Forward and Restoration**  Inspired by stochastic depth (Huang et al., 2016), we introduce a Bernoulli random variable $b_\ell \in \{0, 1\}$ to determine whether the $\ell$-th layer will apply shuffle-based regularization. If $b_\ell = 1$, the input token sequence $X_{\ell-1}$ will be randomly shuffled into $X'_{\ell-1}$, thereby encouraging positional transformation invariance. Otherwise, $X_{\ell-1}$ remains unchanged. We denote this operation by $\pi(\cdot \mid b_\ell)$, and its inverse $\pi^{-1}(\cdot \mid b_\ell)$ restores the shuffled output $X_\ell$ to the original order to avoid recursive effects on later layers:

$$X_\ell = \pi_\ell^{-1}\Big(s_\ell\big(\pi(X_{\ell-1} \mid b_\ell)\big)\Big). \tag{6}$$

**Layer-Wise Probability Assignment**  Additionally, each Vim layer is assigned a distinct probability of applying SLWS, reflecting the intuition that deeper layers should exhibit greater transformation invariance. In this work, we use a linear scheduling function starting with $\ell = 0$. Specifically, the probability $p_\ell$ of applying shuffle regularization at the $\ell$-th layer is:

$$P(b_\ell = 1) = \frac{\ell}{L} P_L, \tag{7}$$

where $P_L$ is a hyperparameter. Since we shuffle tokens according to a discrete uniform distribution, the probability that the $i$-th token moves to the $j$-th position is:

$$\begin{aligned} P\left(x_i^\ell \Rightarrow x_j'^\ell\right) &= \frac{1}{L+1} P(b_\ell = 1) \\ &= \frac{\ell}{(L+1)L} P_L. \end{aligned} \tag{8}$$

Notably, hierarchical Mamba architectures with spatial downsampling operations are incompatible with SLWS, as token sequence length reduction prevents output order restoration. Additionally, SLWS fundamentally differs from random scanning methods. Because our layer-dependent probability allocation imposes progressive regularization intensity that aligns with hierarchical semantic abstraction.

### 3.2.1. EFFICIENCY ANALYSIS

Fig. 1 and Algorithm 1 illustrate SLWS for Vim training with PyTorch pseudo-code. Random index generation incurs $O(L)$ complexity, while sorting for restoration adds $O(L \log L)$. Because we apply the same random index to the entire batch, the batch size does not inflate these costs. Consequently, SLWS introduces only $O(L \log L)$ additional maximal overhead, and our ablation results in Section 4.3 confirm the minimal impact on overall training efficiency.

Overall, SLWS offers several key advantages: (1) It is easy to implement and does not alter the model architecture, adding no extra cost at inference time. (2) It fosters stronger

---

**Algorithm 1** Layer-Wise Shuffle forward

---

**Require:** token sequence $X_{\ell-1} \in \mathbb{R}^{B \times T \times D}$,
    layer $s_\ell$, probability $p_\ell$, training flag $F$
**Ensure:** token sequence $X_\ell$
1:  # this layer is trained with regularization
2:  **if** $F$ and rand(1) $< p_\ell$ **then**
3:     shuffle_indices = randperm(T).expand(B, 1, D)
4:     restore_indices = argsort(shuffle_indices, dim=1)
5:     $X'_{\ell-1}$ = gather($X_{\ell-1}$, 1, shuffle_indices)
6:     $X'_\ell = s_\ell(X'_{\ell-1})$
7:     $X_\ell$ = gather($X'_\ell$, 1, restore_indices)
8:  **else**
9:     # inference or trained without regularization
10:    $X_\ell = s_\ell(X_{\ell-1})$
11:  **end if**
12:  Return: $X_\ell$

---

modeling of 2D visual data by encouraging position invariance in deeper layers. (3) By increasing task complexity, it helps mitigate overfitting without incurring heavy computational overhead in training.

### 3.3. Masked Pre-training for Vim

The fundamental idea of visual masked modeling is to reconstruct the complete target by leveraging relationships between unmasked image patches, thereby capturing complex semantic dependencies. We establish a simple masked pre-training pipeline for the Vim encoder, as formulated in Eq. (5) and illustrated in Fig. 2. Alongside the Vim student encoder, we employ frozen CLIP vision encoders (Radford et al., 2021) as the teacher tokenizers $\mathcal{T}$, which provide feature targets. Inspired by MAE (He et al., 2022), our approach adopts a auto-encoder design, featuring a lightweight self-attention decoder $d$ that reconstructs the Vim features $f(X^v)$ to match the teacher outputs. To enhance training stability, we apply normalization layers to encoded features, decoder outputs, and teacher targets. We further employ the smooth-$\ell_1$ loss for the distance metric dist $[\cdot, \cdot]$.

## 4. Experiments

We conducted extensive experiments to evaluate Vim training, exploring non-hierarchical models trained via supervised classification and pre-training paradigms, assessing their downstream task performance, and performing detailed algorithm analysis through ablation studies. We conduct both horizontal and vertical comparisons to analyze our model and approach.

### 4.1. Implementation Settings

We evaluate various sizes of non-hierarchical Vision Mamba models and details of settings are listed in Appendix A. Configurations of non-hierarchical models with different sizes

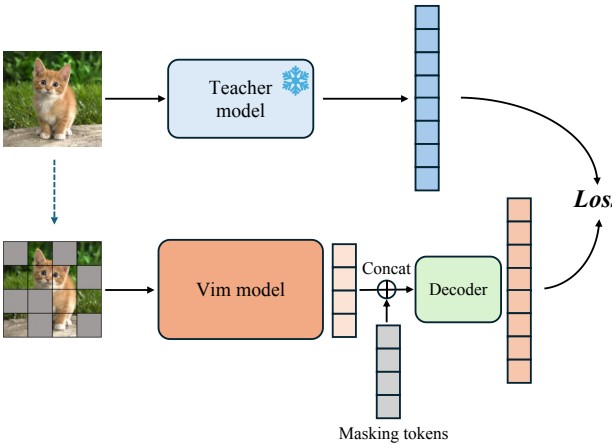

Figure 2: **Masked feature distillation pipeline.** A frozen semantic-rich teacher as tokenizer produces target for the student branch, which is in auto-encoder style.

involved in experiments are listed in the Table 1, in which MambaR (Wang et al., 2024) models add a group of extra tokens based on Vim (Zhu et al., 2024). We use the AdamW optimizer (Loshchilov & Hutter, 2019) with a cosine learning rate schedule and employ BFloat16 precision to enhance training stability. Additionally, we report results using Exponential Moving Average.

Table 1: **Model configurations.**

| Model | Block Config. | Width | Depth | #Param. (M) |
|---|---|---|---|---|
| ViT-B | Attention+MLP | 768 | 12 | 86 |
| Vim-B | Mamba | 768 | 24 | 98 |
| Vim-M | Mamba | 576 | 32 | 74 |
| MambaR-B | Mamba | 768 | 24 | 99 |
| MambaMLP-B | Mamba+MLP | 768 | 12 | 85 |
| ViT-L | Attention+MLP | 1024 | 24 | 309 |
| Vim-L | Mamba | 1024 | 40 | 284 |
| MambaR-L | Mamba | 1024 | 48 | 341 |
| MambaMLP-L | Mamba+MLP | 1024 | 24 | 297 |
| ViT-H | Attention+MLP | 1280 | 32 | 632 |
| MambaMLP-H | Mamba+MLP | 1536 | 24 | 662 |

For supervised training, we train from scratch on ImageNet-1K (Deng et al., 2009), which contains 1.28 million samples for the classification task. Middle and base-size models are trained for 300 epochs with a batch size of 2048, while large models are trained for 200 epochs with a batch size of 1024. The shuffle rate $P_L$ is set to 0.5 for middle and base-size models and 0.6 for large models. Following the VideoMamba (Li et al., 2024a) setup, a [CLS] token is prepended to the token sequences for classification.

For MFD pre-training, we use frozen CLIP vision encoders as tokenizers. Inspired by MAE (He et al., 2022), our decoder is a lightweight self-attention transformer with four blocks and a hidden dimension of 512. We apply layer

Table 2: **Vim training comparisons**, where "S." indicates SLWS, "sup." indicates supervised classification, "reg." refers to token registers (Wang et al., 2024), and "cont." denotes contrastive training. All models are evaluated on the ImageNet-1K benchmark.

| Model | Training tech. | #Params | Epoch | Acc.(%) |
|---|---|---|---|---|
| *supervised* | | | | |
| Vim-M | sup. | 74M | 300 | 80.9 |
| Vim-B | sup. | 98M | 300 | 79.8 |
| Vim-B *[14 stride]* | sup. | 98M | 300 | 81.2 |
| Vim-L | sup. | 284M | 300 | collapsed |
| Vim-M | sup., S. | 74M | 300 | 82.8 *(+1.9)* |
| Vim-B | sup., S. | 98M | 300 | 82.7 *(+2.9)* |
| Vim-L | sup., S. | 284M | 200 | 82.9 |
| Vim-L *[384 res.]* | sup., S. | 284M | 220 | **84.5** |
| MambaR-B | sup., reg. | 99M | 220 | 83.0 |
| MambaMLP-L | sup. | 297M | 300 | 81.4 |
| MambaR-B | sup., reg., S. | 99M | 220 | 83.1 *(+0.1)* |
| MambaMLP-L | sup., S. | 297M | 300 | 82.9 *(+1.5)* |
| *pre-training* | | | | |
| MambaMLP-B | cont. | 85M | 300 | 81.4 |
| MambaMLP-B | MAE | 85M | 300 | 81.6 |
| MambaMLP-B | ARM | 85M | 300 | 82.5 |
| MambaMLP-B | ARM | 85M | 1600 | 83.2 |
| MambaMLP-L | ARM | 297M | 1600 | 84.5 |
| MambaMLP-H | ARM | 662M | 800 | 85.0 |
| MambaMLP-B | MAE, S. | 85M | 300 | 82.0 *(+0.4)* |
| MambaMLP-B | MFD, S. | 85M | 300 | 84.3 *(+1.1)* |
| MambaMLP-L | MFD | 297M | 300 | 86.4 *(+1.9)* |
| MambaMLP-L | MFD, S. | 297M | 300 | 86.7 *(+2.2)* |
| MambaMLP-H | MFD, S. | 662M | 300 | **87.6** *(+2.6)* |

normalization to the output features to improve training stability. During pre-training, we use image sizes of 192 and 224 for the MAE and MFD pipelines, respectively. The shuffle rate is set to 0.4 for large and huge models. For the masking strategy in MFD, we follow existing studies (Peng et al., 2023; Hou et al., 2022) by setting the masking ratio to 0.5 and 0.6 with utilizing attentive masking.

### 4.2. Main Results

**Vim Training Comparison** To evaluate the SLWS regularization and MFD training pipeline, we compare it with state-of-the-art (SOTA) training methods in both supervised and pre-training settings. Table 2 presents the results, including Vim (Zhu et al., 2024), MambaR (Wang et al., 2024), and MambaMLP (Ren et al., 2024). Notably, ARM and MAE pre-train models with an input resolution of $192 \times 192$ and subsequently fine-tune with $224 \times 224$. We utilize a CLIP-B for MambaMLP-B with a CLIP-L for MambaMLP-L and MambaMLP-H as teacher tokenizers, respectively. Based on these results, we draw the following observations:

(1) SLWS significantly improves supervised Vim training across model scales. For the middle-sized Vim-M, SLWS boosts accuracy by 1.9%, and for the base-sized Vim-B, the gain reaches 2.9% (from 79.8% to 82.7%).

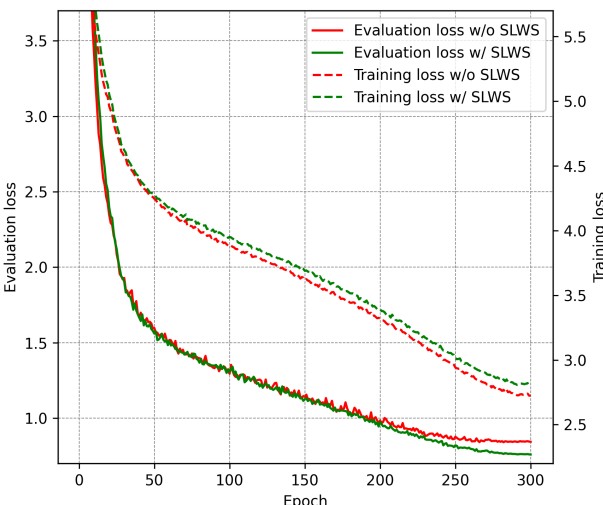

Figure 3: **Training and evaluation loss curves** for 300 epochs middle-size Vims.

Notably, SLWS enables stable training of the previously collapsing Vim-L (284M parameters), achieving 82.9% accuracy and 84.5% with 384×384 fine-tuning.

(2) MFD pre-training substantially enhances Vim capabilities. When combined with SLWS, MambaMLP-B achieves 84.3% accuracy (+1.1% over the ARM baseline), while MambaMLP-L reaches 86.7%, surpassing ARM's 1600-epoch result (84.5%) within just 300 epochs. This demonstrates a clear training efficiency advantage over previous methods and highlights the importance of leveraging a semantic-rich tokenizer.

(3) SLWS provides complementary benefits across training paradigms. In MAE pre-training, SLWS improves MambaMLP-B by 0.4% (81.6% to 82.0%). For MFD, we observe a 0.3% improvement over MambaMLP-L (86.4% to 86.7%), and SLWS enables MambaMLP-H to achieve 87.6%, i.e., the new state-of-the-art result for Vision Mamba on ImageNet-1K.

Consequently, SLWS not only prevents collapse in supervised learning of large models through stochastic regularization *but also* enhances cross-paradigm generalization without any architectural changes. It is also worth noting that combining MFD with SLWS is particularly effective for non-hierarchical Vim training. Beyond the above accuracy evidence for mitigating overfitting, we plot the training and evaluation curves in Fig. 3 for further demonstration. We observe that the model trained with SLWS stabilizes at a higher training loss yet achieves a lower evaluation loss. By contrast, the ablated model tends to overfit, showing a lower training loss but a higher error rate on evaluation. This confirms that SLWS effectively adds perturbation to

Table 3: **ImageNet-1K classification comparison** among different backbone and training methods.

| Model | Training | #Params | FLOPs | Acc.(%) |
|---|---|---|---|---|
| *Hierarchical* | | | | |
| RegNetY-4G | sup. | 21M | 4G | 80.0 |
| RegNetY-8G | sup. | 39M | 8G | 81.7 |
| RegNetY-16G | sup. | 84M | 16G | 82.9 |
| ConvNeXt-T | sup. | 29M | 4.5G | 82.1 |
| ConvNeXt-S | sup. | 50M | 8.7G | 83.1 |
| ConvNeXt-B | sup. | 89M | 15.4G | 83.8 |
| Swin-T | sup. | 28M | 4.6G | 81.3 |
| Swin-S | sup. | 50M | 8.7G | 83.0 |
| Swin-B | sup. | 88M | 15.4G | 83.5 |
| Swin-B | SimMIM | 88M | 15.4G | 84.0 |
| Swin-L | SimMIM | 197M | 35.8G | 85.4 |
| VMamba-T | sup. | 31M | 4.9G | 82.5 |
| VMamba-S | sup. | 50M | 8.7G | 83.6 |
| VMamba-B | sup. | 89M | 15.4G | 83.9 |
| *Non-Hierarchical* | | | | |
| ConvNeXt-S | sup. | 22M | 4.3G | 79.7 |
| ConvNeXt-B | sup. | 87M | 16.9G | 82.0 |
| DeiT-S | sup. | 22M | 4.6G | 79.8 |
| DeiT-B | Distill. | 87M | 17.6G | 81.9 |
| ViT-B *[MAE sup.]* | sup. | 87M | 17.6G | 82.3 |
| ViT-L *[MAE sup.]* | sup. | 309M | 61.6G | 82.6 |
| ViT-B | MAE | 87M | 17.6G | 83.6 |
| ViT-L | MAE | 309M | 61.6G | 85.9 |
| ViT-H *[14 stride]* | MAE | 632M | 167G | 86.9 |
| ViT-B | MaskDistill | 87M | 17.6G | 85.0 |
| ViT-L | MaskDistill | 309M | 61.6G | 87.6 |
| ViT-B | BEITv2 | 87M | 17.6G | 85.0 |
| ViT-L | BEITv2 | 309M | 61.6G | 87.3 |
| Vim-S | sup. | 26M | 4.3G | 80.5 |
| VideoMamba-S | sup. | 26M | 4.3G | 81.2 |
| VideoMamba-M | sup. | 74M | 12.7G | 80.9 |
| VideoMamba-M | self-Distill. | 74M | 12.7G | 82.8 |
| LocalViM-S | sup. | 28M | 4.8G | 81.2 |
| PlainMamba-L2 | sup. | 25M | 8.1G | 81.6 |
| PlainMamba-L3 | sup. | 50M | 14.4G | 82.3 |
| MambaR-S | sup., reg. | 28M | 4.5G | 81.1 |
| MambaR-B | sup., reg. | 99M | 17.8G | 83.0 |
| MambaR-L | sup., reg. | 341M | 64.2G | 83.6 |
| MambaR-L *[384 res.]* | sup., reg. | 341M | 179G | 84.5 |
| MambaMLP-B | ARM | 85M | 15.5G | 83.2 |
| MambaMLP-L | ARM | 297M | 54.7G | 84.5 |
| MambaMLP-H | ARM | 662M | 123G | 85.0 |
| MambaMLP-B | MFD, S. | 85M | 15.5G | 84.3 |
| MambaMLP-L | MFD, S. | 297M | 54.7G | 86.7 |
| MambaMLP-H | MFD, S. | 662M | 123G | 87.6 |

sequential perception, raising task complexity and reducing the overfitting risk for Vim.

**Comparison to Various Backbones.** Table 3 reports ImageNet-1K classification results across a range of backbones. We include CNN-based methods (RegNetY (Radosavovic et al., 2020), ConvNeXt (Liu et al., 2022b)), hierarchical Transformers (Swin (Liu et al., 2021) trained with SimMIM (Xie et al., 2022)), and ViT variants trained with DeiT (Touvron et al., 2021), MAE, MaskDistill (Peng et al., 2023), or BEITv2 (Peng et al., 2022). We also list

SSM-based approaches (VMamba (Liu et al., 2024c), Vim, VideoMamba (Li et al., 2024a), LocalViM (Huang et al., 2024), PlainMamba (Yang et al., 2024a), MambaR (Wang et al., 2024), ARM). Under purely supervised training, several SSM-based models match or exceed the performance of their CNN and hierarchical Transformer counterparts at comparable model sizes. When pre-training is introduced, masked modeling generally boosts performance across architectures. However, there remains a noticeable gap between the previous best SSM-based results and advanced ViT models trained via masked image modeling (e.g., ARM 85.9% vs MAE 86.9%). Moreover, the introduction of our

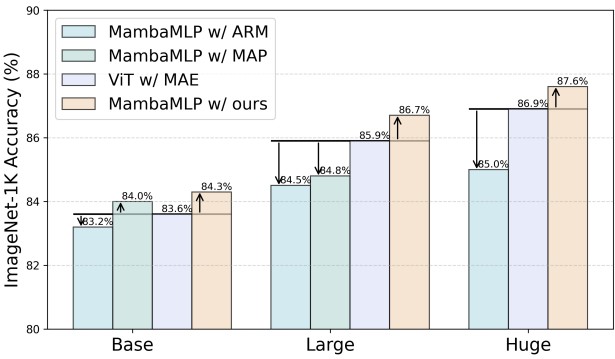

Figure 4: **Pre-training comparison anchored by MAE** on ImageNet-1K.

SLWS + MFD pipeline narrows this gap considerably. By further unlocking the potential of non-hierarchical Vim-like models, it enables them to outperform ViT models trained with MAE, demonstrated by Fig. 4. This substantial improvement underscores the value of our approach in improving the performance of Vim models.

**Dense Prediction Downstream Tasks** To evaluate model capabilities, we conduct semantic segmentation experiments on ADE20K, detection and instance segmentation on COCO2017 benchmark.

For segmentation experiment, we adopt the UPerNet (Xiao et al., 2018) head on ImageNet-1K trained models. All the models are trained for 160K iterations with batch size 16. Following the common settings (Chen et al., 2023; Yang et al., 2024a; Wang et al., 2024), we use an Adam optimizer with 0.01 weight decay and a polynomial learning rate schedule. The learning rates of the base and large-size models are set as 6e-5 and 3e-5, respectively. The [CLS] and register tokens are discarded in the segmentation task. As shown in Table 4, our SLWS-regularized MambaR-B surpasses both ViT-B and its non-SLWS counterpart, which consistently demonstrates the superiority brought by the proposed SLWS regularization. When integrating the multi-scale adapter configuration (Chen et al., 2023), MambaR-

Table 4: **Semantic segmentation results on ADE20K Val.** Computation FLOPs are measured under 512×2048 input resolution. "MS" means multi-scale test.

| Model | #Param. | FLOPs | mIoU | +MS |
|---|---|---|---|---|
| ResNet-50 | 67M | 953G | 42.1 | 42.8 |
| ResNet-101 | 85M | 1030G | 42.9 | 44.0 |
| ConvNeXt-B | 122M | 1170G | 49.1 | 49.9 |
| DeiT-B+MLN | 144M | 2007G | 45.5 | 47.2 |
| ViT-B | 127M | - | 46.1 | 47.1 |
| ViT-Adapter-B | 134M | 632G | 48.8 | 49.7 |
| ViT-L [MAE] | 127M | - | 53.6 | - |
| Swin-B | 121M | 1170G | 48.1 | 49.7 |
| ViM-S | 46M | - | 44.9 | - |
| ViM-B | 131M | 477G | 45.2 | - |
| MambaR-B | 132M | - | 47.7 | - |
| MambaR-L | 377M | - | 49.1 | - |
| Vim-M [S.] | 106M | 384G | 47.2 | 48.2 |
| Vim-B [S.] | 131M | 477G | 47.0 | 48.3 |
| MambaR-B [S.] | 131M | 477G | 48.2 | 48.9 |
| MambaR-Adapter-B [S.] | 145M | 1428G | 49.3 | 50.1 |
| MambaMLP-L [MFD, S.] | 324M | 1270G | **53.8** | - |

Adapter-B outperforms ViT-Adapter-B by 0.5%. Additionally, our MFD+SLWS framework enables MambaMLP-L to match MAE ViT-L's performance.

Table 5: **Object detection and instance segmentation results**. FLOPs are calculated with size 1280×800. Gray fonts indicate the models pre-trained on ImageNet-21K.

| Model | #Param. | FLOPs | $AP^b$ | $AP^b_{50}$ | $AP^b_{75}$ | $Ap^m$ | $AP^m_{50}$ | $Ap^m_{75}$ |
|---|---|---|---|---|---|---|---|---|
| ConvNeXt-B | 108M | 486G | 47.0 | 69.4 | 51.7 | 42.7 | 66.3 | 46.0 |
| Swin-B | 107M | 496G | 46.9 | - | - | 42.3 | - | - |
| ViT-B | 114M | - | 42.9 | 65.7 | 46.8 | 39.4 | 62.6 | 42.0 |
| ViT-L | 337M | - | 45.7 | 68.9 | 49.4 | 41.5 | 65.6 | 44.6 |
| ViT-Adapter-B | 120M | - | 47.0 | 68.2 | 51.4 | 41.8 | 65.1 | 44.9 |
| ViT-Adapter-L | 348M | - | 48.7 | 70.1 | 53.2 | 43.3 | 67.0 | 46.9 |
| PlainMamba-L3 | 79M | 696G | 46.8 | 68 | 51.1 | 41.2 | 64.7 | 43.9 |
| Vim-M [S.] | 103M | 564G | 46.8 | 68.8 | 50.7 | 41.8 | 65.6 | 44.8 |
| MambaR-B [S.] | 131M | 726G | **47.7** | **69.7** | **51.8** | **42.6** | **66.7** | **45.8** |
| MambaR-L [S.] | 383M | 1734G | **48.9** | **70.8** | **53.4** | **43.6** | **67.4** | **47.0** |

For downstream object detection and instance segmentation tasks, we follow previous work to evaluate our method. The Mask R-CNN (He et al., 2017) structure is adopted with 1× schedule for 12-epoch fine-tuning. We utilize the commonly adopted settings in previous work (Liu et al., 2021) and compare to different-type backbones. To compute the multi-scale features to fit the FPN network structure, we use the Adapter setup following (Yang et al., 2024a; Chen et al., 2023). The results are reported in Table 5. It can be seen that our middle-size model is on par with the corresponding CNN and Transformer model, while the base-size model traine with registers and SLWS outperforms ViT-Adapter-B and ConvNext-B by 0.7 points $AP^b$. MambaR-L demonstrates higher $AP^b$/$AP^m$ and even outperforms ImageNet-21K pretrained ViT-Adapter-L and ViT-L.

## 4.3. Ablation Studies

In this subsection, we perform ablation studies by varying the settings of the SLWS regularization to investigate its effects and provide an in-depth analysis. We use middle-size vanilla Vim models as the default for experiments.

Table 6: **Ablation study on training throughput.** Higher throughput (images/second) is better under the same setting.

| Setting | 128 | 224 | 384 | 512 | 768 |
|---|---|---|---|---|---|
| w/o SLWS. | 315.7 | 167.9 | 56.8 | 29.0 | 13.97 |
| w/ SLWS. | 311.4 | 164.8 | 55.7 | 28.6 | 13.72 |
| Loss (%) ↓ | 1.36 | 1.85 | 1.94 | 1.38 | 1.79 |

**SLWS has a Negligible Impact on Training Throughput.** SLWS operates on both input and output sequences of Mamba blocks, with efficiency analysis detailed in Section 3.2. To empirically evaluate its computational overhead, we conduct throughput measurements using standard image resolutions ranging from 128×128 to 768×768. Results in Table 6 demonstrate consistent throughput reduction below 2% across all resolutions. This negligible overhead confirms SLWS as a simple but effective training regularization technique for vision mamba models.

Table 7: **Ablation studies on probability settings.** "D" and "E" denote decreased linear, and exponential probability assignments, respectively.

| Type | $P_L$ | Acc. | Type | $P_L$ | Acc. |
|---|---|---|---|---|---|
| | 0.4 | 82.3 | Layer-depend. (D) | 0.5 | 81.2 |
| | 0.5 | 82.7 | Layer-depend. (E) | 0.5 | 82.2 |
| Layer-depend. | 0.6 | 82.4 | | 0.1 | 81.5 |
| | 0.7 | 82.4 | Constant | 0.4 | 81.1 |

**Layer-Wise Probability Assignment is Necessary.** The layer-wise probability is a crucial component of the SLWS design, introducing a semantic level prior across different layers. Table 7 presents the results under various probability assignment settings. Since shallower blocks are more sensitive to patch positions, the layer-dependent cases generally outperform the constant settings. We also provide a decreased linear probability assignment comparison, which takes larger shuffle probabilities for shallower layers. The result further demonstrates the correctness of the semantic level prior. Besides the linear setting, we experiment with a exponential one, i.e. a modification of Eq. (7) to $P(b_\ell = 1) = P_L^{(L-\ell+1)}$, which has a performance drop of 0.5 points compared to the vanilla linear one.

**Including the `[CLS]` Token in Shuffling Slightly Improves Performance.** As the `[CLS]` token is used for

Table 8: **Ablation studies on [CLS] token shuffling** with different size of models.

| [CLS] token in Shuffling | Middle | Base | Large |
|---|---|---|---|
| × | 82.6 | 82.6 | 82.8 |
| ✓ | 82.7 | 82.6 | 82.9 |

supervised classification training except for MambaR configuration, we investigate whether including it in the shuffling process affects performance. The ablation results for different model sizes on ImageNet-1K are shown in Table 8. We observe that including the `[CLS]` token in shuffling yields slightly better performance under the same settings for middle and large models. Consequently, for code simplicity, we shuffle the entire sequence by default, and the same approach applies when using registers.

## 5. Conclusion

We present Stochastic Layer-Wise Shuffle, a specialized regularization method for non-hierarchical Vision Mamba training that addresses overfitting through layer-dependent sequence perturbations. By progressively increasing shuffle probabilities across layers, SLWS enhances positional transformation invariance in deeper semantic abstractions while preserving low-level spatial sensitivity. This approach achieves significant improvements for supervised training of Vision Mamba. When integrated with masked feature distillation, our Vim models establish new state-of-the-art results on ImageNet-1K and dense prediction tasks among the same type models. The method does not introduce architecture modification and has negligible overhead, effectively unlocking the potential of Vision Mamba models.

## Acknowledgement

Thanks Di Yang from SII for his help. This work is supported by the National Key R&D Program of China (No. 2022ZD0160900), Jiangsu Frontier Technology Research and Development Program (No. BF2024076), and the Collaborative Innovation Center of Novel Software Technology and Industrialization. This work is funded by Nanjing University-China Mobile Communications Group Co.,Ltd. Joint Institute.

## Impact Statement

This paper presents work whose goal is to advance the field of Deep Learning. There are many potential societal consequences of our work, none of which we feel must be specifically highlighted here.

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

# A. Implementation Details

## A.1. Supervised Training Settings

In supervised classification training, we build our pipeline with the codebase of DeiT (Touvron et al., 2021) and the setups are listed in the following Table A.1. For MambaMLP-L training, the layer-wise shuffle rate and drop path rate are 0.5. For MambaR, we follow the training setup from Wang et al. (2024), which employs a three-stage strategy equivalent to approximately 220 epochs of training at an input resolution of 224 except setting layer-wise shuffle rate to 0.1.

Table A.1: Supervised training implementation settings.

| Config | Base & Middle | Large |
|---|---|---|
| optimizer | AdamW | |
| base learning rate | 5e-4 | |
| weight decay | 0.1 | 0.15 |
| layer-wise lr decay | 0 | |
| learning rate schedule | cosine decay | |
| batch size | 2048 | 1024 |
| warmup epochs | 30 | |
| training epochs | 300 | 200 |
| augmentation | RandAug (9, 0.5) | |
| label smoothing | 0.1 | |
| mixup | 0.8 | |
| cutmix | 1 | |
| reprob | 0.25 | |
| drop path rate | 0.5 | 0.7 |
| layer-wise shuffle rate | 0.5 | 0.6 |
| EMA decay | 0.99992 | 0.99992 |

## A.2. Pre-training Settings

We provide the configurations of models used in MFD pre-training in the following Table A.2. During pre-training, we use image sizes of 192 and 224 for the MAE and MFD pipelines, respectively, and the masking ratio for MAE is 0.7. The 224 resolution is a common training setting adopted by works like Vim(Zhu et al., 2024), ViT(Dosovitskiy et al., 2021), and MAE(He et al., 2022). ARM(Ren et al., 2024) uses 192 due to its unique design requirement of dividing images into multiple 64×64 patch groups. To ensure fair comparison under the same training epochs, their MAE experiment also followed this resolution. For MFD pre-training, we used the standard 224 resolution but with only 18.75%-37.5% of ARM's training epochs, significantly reducing computational costs.

Table A.2: Pre-training training implementation settings.

| Config | Base | Large | Huge |
|---|---|---|---|
| optimizer | AdamW | | |
| base learning rate | 1.5e-4 | | |
| weight decay | 0.05 | | |
| learning rate schedule | cosine decay | | |
| batch size | 2048 | 1024 | 1024 |
| warmup epochs | 30 | | |
| training epochs | 300 | 300 | 300 |
| shuffle rate | 0.1 | 0.4 | 0.6 |
| masking ratio | 0.5 | 0.6 | 0.6 |
| augmentation | RandomResizedCrop | | |

