# OpenReview forum: "Stochastic Layer-Wise Shuffle for Improving Vision Mamba Training"
_ICML.cc/2025/Conference — ICML 2025 poster_

### Official Review · Reviewer_bypj · 2025-03-08

**Overall Recommendation:** 3

**Summary:**

This paper proposes a plug-and-play training strategy for Vision Mamba. It shuffers the sequence order of the input tokens layer by layer. The authors conduct mask feature distillation to pre-train the vision Mamba with the proposed layerwise shuffling strategy. Experiments on classification and dense prediction tasks show improvement. The overall writing is easy to understand, and the method is simple and clean.

**Claims And Evidence:**

Yes. The author conducts extensive experiments to validate the method. Several baseline methods， such as Vim and MambaMLP, are adopted to test the proposed method.

**Essential References Not Discussed:**

None

**Experimental Designs Or Analyses:**

Yes, the experimental design follows the masked feature distillation pipeline.

**Methods And Evaluation Criteria:**

Yes, the IoU and accuracy on classification detection and segmentation tasks are reported.

**Other Comments Or Suggestions:**

Please address the concerns in weakness.

**Other Strengths And Weaknesses:**

Strengths:

1. The method is technically sound and well-motivated. Only shuffling the token order is simple and effective, which can impact more vision Mamba works.

2. The experiments are extensive, including pretraining, classification, and dense prediction tasks.

3. The performance improvement shows the effectiveness of the method.

Weakness.

1. The shuffling strategy is only applied to plain architecture. While most vision Mamba have hierarchical structures such as VMamba [1], the authors are suggested to apply it to hierarchical structures. Otherwise, the impact of this work is constrained.

2. The layerwise probability design seems to be handcrafted. Intuitive reasons are not enough to explain it. Please provide more solid reasons or develop a more dedicated design for the layerwise probability.

3. The effects of applying a shuffling strategy directly instead of masked feature distillation should be explored in more detail since it is the most straightforward way to apply the proposed shuffling strategy. For example, is it possible to apply the shuffling strategy to the direct pretraining of image classification or MAE-style [2] pretraining?  Since no teacher model weights are used, it can be more convenient for other followers.

[1] VMamba: Visual State Space Model in NeurIPS24. https://arxiv.org/pdf/2401.10166

[2] Masked Autoencoders Are Scalable Vision Learners in CVPR22. https://arxiv.org/abs/2111.06377

**Questions For Authors:**

None

**Relation To Broader Scientific Literature:**

The method could impact the pretraining of current vision mamba based method.

**Theoretical Claims:**

Yes, the method is simple and easy to understand. A shuffling strategy is applied to Mamba for pertaining.

---

> ### Author Rebuttal · Authors · 2025-04-01
>
> Thanks for your constructive comments.  We are glad that you found our work is technically sound and well-motivated, has extensive experiments with effectiveness.  We provide our feedback as follows.
>
> > **Q1: Applying SLWS to hierarchical structures.**
>
> **A1:** Hierarchical architectures like VMamba employs a more complex downsampling process, which demonstrates strong performance under Tiny, Small, and Base model sizes. However, there is currentl*y no evidence to confirm its scalability to larger model sizes beyond the Base size*. Furthermore, the hierarchical structure of VMamba is incompatible with our proposed straightforward shuffle-based regularization method. This is because downsampling at certain layers leads to inconsistent input-output sequence lengths, making it infeasible to implement the "order restoration" step in our method. This limitation is discussed in lines 253–260 of the submitted manuscript.
>
> In contrast, plain vision Mamba models feature a non-hierarchical architecture. **1)** This design is simpler, more fundamental, easier to stack, **2)** and seamlessly compatible with a wide range of existing sequence modeling frameworks like MLLMs, as well as training paradigm like MAE and masked feature distillation. **3)** Such architectures (e.g., Vim, Mamba-R, ARM, PlainMamba) have been widely adopted and benchmarked in prior research. Building upon this foundation, we propose SLWS, a plug-and-play regularization method that further enhances the scalability of these models. Our approach achieves state-of-the-art results on ImageNet among Vision Mamba variants, contributing to the exploration of efficient regularization strategies for plain Mamba architectures.
>
> > **Q2: Linear layer-wise probability design reasons**
>
> **A2:** The layer-wise probability design is inspired by well-established practices in hierarchical feature learning. For instance, Stochastic Depth[1] adopts a linear probability schedule across layers, guided by the intuition that deeper layers handle higher-level semantics and are more robust to structural variations. Similarly, our linear probability design aligns with the inherent hierarchy of visual processing. To assuage reviewers' concern, we provide a further dedicated design for the layerwise probability experiment here (i.e., $p_{\ell}=P_L^{(L-\ell+1)}$, similar to the *layer-wise learning rate decay* form, where $P_L=0.5$) and the corresponding test accuracy is 82.2, which is lower than the original linear setting of 82.7 in Table 5. We think that it is due to the fact that the *power function form* of the design leads to too low a probability for the middle layers, leading to inadequate regularization.
>
> > **Q3: Applying SLWS to diverse training paradigms and their horizontal and vertical comparisons.**
>
> **A3:** As demonstrated in Table 1, we conducted comprehensive experiments to validate the effectiveness of the SLWS across non-hierarchical Mamba architectures and diverse training paradigms, including:  naive supervised classification training, self-supervised MAE pretraining,  and Masked feature distillation (MFD). These results and common improvements confirm that SLWS can be compatible with the  training paradigms. For example, SLWS even elevates the accuracy of the previously collapse-prone Vim-L model to a 84.5%, and applying the shuffling strategy to the MAE-style pretraining brings a 0.4 points gains.
>
> ------
>
> Thank you for all your comments, which we believe have strengthened our work; we hope our responses have addressed your remaining concerns.
>
> [1] Deep networks with stochastic depth, ECCV'16.

---

> > ### Comment · Reviewer_bypj · 2025-04-07
> >
> > I keep the original rating after reading the reviews and rebuttal.

---

### Official Review · Reviewer_knen · 2025-03-14

**Overall Recommendation:** 3

**Summary:**

This work introduces a stochastic hierarchical shuffle strategy SLWS for Vision Mamba (Vim) that successfully solves the overfitting issue of Mamba models in large-scale datasets without changing the model architecture, effectively improving the training of Vim. According to this paper, SLWS can help Vim achieve leading performance on the officially recognized ImageNet-1K without introducing significant additional computational overhead.

**Claims And Evidence:**

Yes, this paper provides the corresponding computational analysis and experimental proof for SLWS.

**Essential References Not Discussed:**

No.

**Experimental Designs Or Analyses:**

The effectiveness of SLWS in solving the overfitting problem of Mamba model is demonstrated in Figure 2 by analyzing the training loss and evaluation loss with and without SLWS.

**Methods And Evaluation Criteria:**

Yes, this work effectively improves the overfitting problem of Mamba models in large-scale datasets.

**Other Comments Or Suggestions:**

1. The LWS in the legend of Figure 2 is not specified in the text as to whether it refers to a SLWS .
2. Related work vision backbone example of the reference work is a bit out of date.

**Other Strengths And Weaknesses:**

Strengths.
1. The proposed SLWS significantly improves the performance of non-hierarchical Mamba models on large-scale datasets.
2. SLWS does not incur significant computational overhead. 3. Numerous experiments have demonstrated the effectiveness of SLWS.


Weakness.
1. Token shuffle may break some connections between tokens that are supposed to be visually linked, and how SLWS avoids it.
2. During pre-training, this work uses image sizes of 192 and 224 for the MAE and MFD pipelines, respectively. Why different pre-training resolution settings are used for ARM, MAE and MFD. Could this be the reason for the difference in performance between MAE, ARM and MFD.
3. We observe that VMamba with similar parameter counts has higher performance compared to Vim paired with the SLWS strategy (83.9 vs 82.7). Whether hierarchical visual Mamba design is more effective in mitigating the overfitting problem of Mamba model in large-scale dataset.
4. The improvement of SLWS for MambaR looks limited in Table 1, what is the reason for this.
5. The MAE and ARM comparison setups pre-trained from scratch seem to be inadequate when compared to MFDs that have introduced pre-trained CLIP distillation. How does MFD compare to some distillation pre-training methods, such as MaskDistill paired with MambaMLP results.
6. The ablation in Table 5 only looks to demonstrate that a dynamic shuffle strategy is better than a constant one, and is not sufficient to justify the proposed “realization of stronger semantic awareness in deeper layers requires translational invariance of patch locations, whereas shallower layers must remain location-sensitive”. What would be the effect on the results if a larger PL was used for shallower layers and a smaller PL for deeper layers.

**Questions For Authors:**

Although SLWS clearly improves the performance of non-hierarchical Mamba models on large-scale datasets, some key experiments and some explanations (see weakness) are needed. I suggest that the authors provide more experimental evidence to further support their work.

**Relation To Broader Scientific Literature:**

[1] similarly discusses the impact of a training strategy for learning token location invariance on the model and demonstrates its effectiveness.
References: [1] Efficient Training of Visual Transformers with Small Datasets

**Theoretical Claims:**

Yes, This work asserts that SLWS possesses negligible computational overhead, and the efficiency analysis in Section 3.2.1 and Table 4 provide the corresponding proof. This paper claims that SLWS can solve the overfitting problem of the Mamba model on large datasets, and Table 1 and Figure 2 provide the corresponding experimental proofs.

---

> ### Author Rebuttal · Authors · 2025-04-01
>
> Thanks for your constructive comments.  We are glad that you found our work significantly improves the performance of non-hierarchical Mamba models and does not incur significant computational overhead.  We provide our feedback as follows.
>
> > **Q1: SLWS's shuffle effects for visual connections between tokens**.
>
> **A1:** Mamba's scanning mechanisms inherently restrict token interactions to 1-D adjacency within the sequence, which is misaligned with the 2-D structural priors of images. Our shuffle operation introduces randomness to enable non-local token interactions as regularization, effectively sampling from the full \($O(n^2)$\) dependency space while preserving efficiency. Critically, we implement three safeguards to maintain positional coherence:  **1)** Positional encodings explicitly retain the inherent locality of the original image data.  **2)** Layer-wise shuffle probability: Deeper layers (handling global features) adopt higher shuffle probabilities, while shallow layers (processing low-level information) is more likely to remain unshuffled.  **3)** Order restoration: The original sequence order is restored after each shuffled layer, preventing recursive disruption of positional relationships. **4)** Experimental results demonstrate  both the rationality and effectiveness of SLWS.
>
> > **Q2: Training resolution setting and performance difference between training pipelines.**
>
> **A2:** The 224 resolution is a common training setting adopted by works like Vim, ViT, and MAE. ARM uses 192 due to its unique design requirement of dividing images into multiple 64×64 patch groups. To ensure fair comparison *under the same training epochs*, their MAE experiment also followed this resolution. For MFD pre-training, we used the standard 224 resolution but with only *18.75%-37.5% of ARM's training epochs*, significantly reducing computational costs. Meanwhile, MAE was trained with the same epochs as that in ARM. Thus, the comparisons are fair, and our method consistently achieves improvements across different training strategies.
>
> >  **Q3: Hierarchical VMamba's effect in mitigating overfitting.**
>
> **A3:** VMamba employs a more sophisticated downsampling process and adopts a hierarchical structure, which demonstrates strong performance in smaller model scales (e.g., Tiny, Small, and Base). However, there is no evidence to suggest that this design inherently benefits training at larger scales, as VMamba has only been extended up to the Base size. Further investigation would be required to validate its effectiveness in mitigating overfitting for larger models. Please also see details in our response to Reviewer mf21 (response 1 & 2).
>
> > **Q4: Performance improvement based on MambaR.**
>
> **A4:**  Mamba-R successfully scales Mamba to the Large size under supervised training but requires the *addition of extra register tokens to the original architecture*, which brings overhead in training and inference. In contrast, SLWS achieves comparable results without any architectural modifications and overhead in inference. Our experiments on Mamba-R aim to demonstrate compatibility with prior techniques, and SLWS still outperforms it by 0.5 points in segmentation tasks, further validating its effectiveness.
>
> > **Q5: Horizontal and vertical comparisons of MAE, ARM and MaskDistill.**
>
> **A5:** Our comparisons encompass both MAE and MFD training *with and without SLWS regularization*, and the results demonstrate that *SLWS consistently improved performance under both training paradigms*. When evaluating cross-strategy performance, MFD indeed benefits from CLIP’s rich semantic knowledge, outperforming MAE. Furthermore, as shown in Table 2, existing self-supervised methods (e.g., ARM’s MambaMLP-L at 84.5) lagged significantly behind MaskDistill’s ViT-L (87.6). However, our MFD-trained MambaMLP-L achieves 86.7, which narrows the gap with MaskDistill’s ViT-L (87.6), showcasing substantial progress.
>
> > Q6: Reversed layer-wise probability experiment for SLWS's semantic awareness priori.
>
> **A6:** Following your suggestion, we conducted experiments with a reversed layer-wise probability strategy (assigning larger shuffle probabilities to shallower layers and smaller ones to deeper layers). The results showed a performance drop of 1.5 points on ImageNet as the table below. Thus, our original statement about semantic-awareness still holds.
>
> | probability config. | constant | layer-wise | reversed layer-wise |
> | ------------------- | -------- | ---------- | ------------------- |
> | **Acc.**            | 81.1     | 82.7       | 81.2                |
>
> > **Suggestions about Figure 2 and related work.**
>
> **A:** Thank for your suggestions and we have revised the legend of Figure 2, as well as included discussed some new related vision backbone literatures like Mamba Vision[CVPR'25], TransNeXt[CVPR'24].
>
> ------
>
> Thank you for all your comments, which we believe have strengthened our work; we hope our responses have addressed your remaining concerns.

---

> > ### Comment · Reviewer_knen · 2025-04-08
> >
> > I have read the rebuttal. My concerns are mostly addressed especially for the fairness and effectiveness of proposed method

---

> > > ### Author Response · Authors · 2025-04-09
> > >
> > > Thank you for your reply and the efforts you have put into the review process, which we believe has been very helpful. We are glad to see that your concerns have been mostly addressed, and we would like to ask if you would consider raising the score further. Thank you again!

---

### Official Review · Reviewer_6YiU · 2025-03-18

**Overall Recommendation:** 3

**Summary:**

This paper introduces a method that addresses overfitting issues when scaling up vanilla Vision Mamba models to larger sizes. The key contribution is a Stochastic Layer-Wise Shuffle (SLWS) regularization technique that randomly shuffles token positions during training with layer-dependent probabilities. Experiments also improvements on classification, detection and segmentation tasks.

**Claims And Evidence:**

The evidence partially supports the paper's claims but has several inconsistencies:

1. The claim that it outperforms similarly-sized models is only selectively supported. When compared to Mamba-Reg models, the improvements are marginal (MambaR-B: 83.0% vs. MambaR-B with SLWS: 83.1%).
2. The claim that vanilla Vision Mamba models couldn't previously be scaled up is contradicted by cited work, including Mamba-R and ARM, which have successfully scaled Vision Mamba to large and even huge sizes.

**Essential References Not Discussed:**

No

**Experimental Designs Or Analyses:**

Yes.

**Methods And Evaluation Criteria:**

Yes

**Other Comments Or Suggestions:**

None.

**Other Strengths And Weaknesses:**

Strengths:
1. Clear motivation and well-written presentation
2. Simple, plug-and-play approach with no inference overhead
3. Comprehensive evaluation across multiple vision tasks

Weaknesses:

1. Limited performance improvements compared to baseline models especially Mamba-R
2. Concerns about disrupting inherent locality of image data through token shuffling
3. Potentially harmful to dense prediction tasks where positional information is crucial

**Questions For Authors:**

1. Can SLWS apply to hierarchical variants?
2. Would other shuffling strategies with locality help?
3. Can you apply SLWS on vision transformer if the claims hold?

**Relation To Broader Scientific Literature:**

1. Recent work in Vision Mamba scaling, particularly Mamba-R [1] and ARM [2], which have successfully scaled the Mamba up.
1. The relationship to positional invariance in vision models and how token shuffling affects spatial understanding.

**Theoretical Claims:**

No theoretical claim.

---

> ### Author Rebuttal · Authors · 2025-04-01
>
> Thanks for your constructive comments.  We are glad that you found our work has clear motivation and comprehensive evaluation, and simple plug-and-play design.  We provide our feedback as follows.
>
> > **Q1: Performance improvements of our models compared to baselines especially Mamba-R**
>
> **A1:** Our proposed SLWS regularization demonstrates significant improvements across both supervised and self-supervised training paradigms. For example:
>
> - It enables successful supervised training of the previously collapse-prone vanilla Vim-L model without architectural modifications, achieving a accuracy of 84.5% on ImageNet.
> - The MambaMLP-H model trained with SLWS reaches 87.5% accuracy, setting a new state-of-the-art result for vision Mamba variants.
>
> These advancements have been acknowledged by reviewers, who highlighted the "impressive results," "measurable improvements," and "significant performance gains" in their feedback.
>
> Regarding Mamba-R, it is an existing method that successfully scales Mamba to Large size under supervised training but *requires the addition of extra register tokens* to the original architecture. In contrast, SLWS achieves comparable results without any architectural modifications. Our experiments on Mamba-R aim to demonstrate compatibility with prior techniques, and SLWS still outperforms it by 0.5 points in segmentation tasks, further validating its effectiveness.
>
> > **Q2 & Q3: Concerns about disrupting inherent locality and positional information of image data and dense prediction**
>
> **A2 & A3:** Mamba's scanning mechanisms inherently restrict token interactions to 1-D adjacency within the sequence, which is misaligned with the 2-D structural priors of images. Our shuffle operation introduces randomness then enables some global token interactions, effectively sampling from the full \($O(n^2)$\) dependency space while preserving efficiency. Critically, we implement three safeguards to maintain positional coherence:  **1)** Positional encodings explicitly retain the inherent locality of the original image data.  **2)** Layer-wise shuffle probability: Deeper layers (handling global features) adopt higher shuffle probabilities, while shallow layers (processing low-level information) remain unshuffled.  **3)** Order restoration: The original sequence order is restored after each shuffled layer, preventing recursive disruption of positional relationships.
>
> For dense prediction tasks (e.g., segmentation), SLWS is used for backbone pre-training and achieves improved performance as listed in Table 3, demonstrating  both the rationality and effectiveness of SLWS.
>
> > **Q4 & Q5: Applying SLWS to hierarchical variants and ViT**
>
> **A4 & A5:**
>
> - Application to Hierarchical Variants:
>   Hierarchical architectures (e.g., VMamba) incorporate downsampling layers, which alter feature map dimensions across stages. This disrupts the sequence length consistency required for SLWS's order restoration step, a critical process for SLWS.  Please also see details in lines 253–260 of the manuscript and our response to Reviewer mf21 (response 1&2).
>
> - Application to Vision Transformers (ViTs):
>   ViTs inherently perform global \($O(n^2)$\) interactions via self-attention. Token shuffling in ViTs would not meaningfully alter their ability to model long-range dependencies. As self-attention is permutation-equivariant, shuffling input tokens does not change the attention output.
>
> > **Q6: Shuffling strategies with locality**
>
> **A6**：Other shuffling strategies with localization might be useful, such as implementation within a certain window. However, this will significantly increase the complexity of the implementation, and we believe that layer-wise probability settings are also necessary, as this is important for conforming to the deep model semantic hierarchy prior.
>
> > **Q7 (review content in Claims And Evidence): "The claim that vanilla Vision Mamba models couldn't previously be scaled up"**
>
> **A7**：We appreciate the reviewer’s attention to this point. But our manuscript does not have such claim. Instead, we explicitly acknowledge in the Introduction  (lines 36–38) and Related Work sections that *"a limited number of ... strategies [ARM, MambaR, MAP] have successfully trained and scaled certain Mamba-based models to Huge sizes."*
>
> ------
>
> Thank you for all your comments, which we believe have strengthened our work; we hope our responses have addressed your remaining concerns.

---

### Official Review · Reviewer_8tDk · 2025-03-18

**Overall Recommendation:** 3

**Summary:**

The paper introduces Stochastic Layer-Wise Shuffle (SLWS), a method designed to enhance the training of Vision Mamba models (ViM). This approach involves applying stochastic shuffling to input tokens at each layer, with the probability of shuffling systematically increasing as the layer depth progresses. Though conceptually simple, SLWS demonstrates significant benefits: it mitigates overfitting, promotes positional invariance across successive blocks, and improves model robustness. These advantages translate to measurable performance gains in both supervised and unsupervised pre-training paradigms.

**Claims And Evidence:**

1. Potential for Stronger Empirical Support on Overfitting Mitigation
SLWS is presented as an effective method for mitigating overfitting in supervised training while maintaining computational efficiency and architectural simplicity. While the paper references extensive experiments to support this claim, the direct empirical evidence—limited to two learning curves—shows only modest reductions in the training-validation accuracy gap. To further substantiate the connection between SLWS and improved generalization, the authors could enhance this section with additional metrics (e.g., validation loss trends across epochs, comparisons of parameter sensitivity) or targeted ablation studies. Such additions would help clarify how the stochastic shuffling mechanism specifically contributes to reducing overfitting.

2. Opportunities for Qualitative Insights in Downstream Tasks
The improved performance of SLWS in semantic segmentation and other downstream tasks is numerically compelling. However, the analysis could be enriched by qualitative examples illustrating how the method fosters robust or positionally invariant features. For instance, visualizations of feature activation patterns (e.g., attention maps or segmentation boundaries) in SLWS-trained models versus baselines could offer intuitive insights into why the method succeeds. While deeper mechanistic analysis might extend beyond the scope of this work, even simple visual comparisons would strengthen the paper’s narrative and provide readers with a clearer understanding of SLWS’s practical benefits.

**Essential References Not Discussed:**

N/A

**Experimental Designs Or Analyses:**

1. Supervised Classification Improvements and Comparison Scope
The paper demonstrates consistent, albeit modest, improvements in supervised classification tasks when applying SLWS, with accuracy gains ranging from 0.4% to 2.9% across Vision Mamba (ViM) and MambaMLP variants. These gains are further shown to generalize across different backbones and training methodologies. However, the analysis primarily evaluates Mamba-based models trained with Masked Feature Distillation (MFD), a specific regularization strategy. While the results are promising, it would be valuable to clarify whether SLWS’s benefits are intrinsic to the method itself or partially reliant on MFD’s regularization effects. Including comparisons with non-MFD-trained baselines (e.g., vanilla ViM or non-Mamba architectures) could help establish SLWS’s broader applicability and ensure equitable comparisons with other training regimes.

2. Clarity in Segmentation Results
In semantic segmentation experiments, the performance gains attributed to SLWS are presented in a table that lacks direct counterparts for certain configurations. For example, the ViM-M models trained with SLWS do not appear to have equivalent baselines without SLWS. This omission complicates efforts to isolate the method’s contribution from architectural or training-specific factors. Providing explicit comparisons for all configurations—or transparently acknowledging these limitations—would enhance interpretability and strengthen confidence in SLWS’s role in the observed improvements.

**Methods And Evaluation Criteria:**

yes.

**Other Comments Or Suggestions:**

* Table 4: Clarify the units, indicate if less/more is better.

**Other Strengths And Weaknesses:**

The proposed Stochastic Layer-Wise Shuffle (SLWS) stands out for its conceptual simplicity and computational efficiency, requiring no architectural modifications while delivering consistent performance gains. The method demonstrates measurable improvements across diverse tasks, including supervised classification (with accuracy increases of 0.4% to 2.9%) and semantic segmentation, suggesting broad applicability. Its ability to enhance positional invariance and reduce overfitting—even under varied backbones and training regimes—further underscores its potential as a versatile training aid.

Weaknesses:
While the empirical results are promising, the paper’s claims would benefit from additional supporting evidence to solidify mechanistic insights. For instance:

Data granularity: Including learning curves (e.g., training vs. validation loss/accuracy trends) would help visualize how SLWS mitigates overfitting in practice.

Qualitative examples: Visualizations of feature activations or segmentation outputs could clarify how SLWS fosters robustness or invariance.

Interpretability of comparisons: Some results tables (e.g., semantic segmentation) lack direct SLWS vs. non-SLWS comparisons for key configurations (e.g., ViM-M), making it challenging to isolate the method’s impact. A brief discussion justifying the presentation format or addressing potential fairness concerns (e.g., MFD-focused evaluations) would enhance methodological transparency.

**Questions For Authors:**

* Would it be possible to provide more evidence about the help of the method to address overfitting in Mamba models?

**Relation To Broader Scientific Literature:**

The paper introduces the shuffling for vim models. It shows yet another transformation to encourage model robust and invariant representation.  Scaling the models has been problematic due to overfitting and other constrains during training.

**Theoretical Claims:**

The algorithm 1 for LWS forward is clear and correct.

---

> ### Author Rebuttal · Authors · 2025-04-01
>
> Thanks for your constructive comments.  We are glad that you found our work is simple but demonstrates significant benefits, and has measurable performance gains across  diverse tasks and training paradigms.  We provide our feedback as follows.
>
> > **Q1-1: Supporting evidence of  learning curves for mitigating overfitting of SLWS**.
>
> **A1-1:** We are glad to indicate that the Figure 2 in our submitted manuscript is exactly what you need, i.e., the training and validation loss curves with and without using SLWS. As the analysis stated in line 318-324 in the right column of the paper, the model trained with SLWS stabilizes at a higher training loss yet achieves a lower evaluation loss and a better final accuracy, which implies the effectiveness of mitigating overfitting. This type of curve are also adopted by some classical work like ResNet and Stochastic Depth[1].
>
> > **Q1-2: Qualitative examples for SLWS**.
>
> **A1-2:** Qualitative visual analysis could help better understand the training strategy. However, since Mamba lacks attention scores like those in ViT, such visualizations are inherently challenging. To enable effective qualitative analysis and intuitive insights, we provide some examples of segmentation outputs comparisons in the anonymous link [https://postimg.cc/GHSPq9Py ]. These visualizations reveal that SLWS pre-trained model achieves more accurate segmentation boundaries when transferred to segmentation task, indicating that SLWS's higher semantic awareness, which is consistent with the quantitative results.
>
> > **Q2: Direct segmentation comparisons of SLWS vs. non-SLWS**
>
> **A2:**  The Vim-M model already exhibited significant disadvantages in classification tasks under non-SLWS training (80.9 vs 82.8), which led us to exclude it from downstream segmentation comparisons. In Table 3, the SLWS-trained MambaR-B achieves a 0.5 higher mIoU when *directly compared to its  non-SLWS baseline*. To further address reviewer concerns, we conducted additional experiments on direct comparison of ViM-B  in the table below, demonstrating that non-SLWS-trained models consistently underperform their SLWS-trained counterparts in segmentation tasks. This further validates the performance generalizability and robustness between SLWS classification training and its downstream tasks.
>
> |     metric\model| SLWS | non-SLWS |
> | --------- | ---- | -------- |
> | Cls. acc. | 82.7 | 81.2     |
> | Seg. mIoU | 47.0 | 45.2    |
>
> > **Q3 (review content in Experimental Designs Or Analyses): SLWS’s gains and relationship to training paradigm**
>
> **A3:** Our study provides comprehensive experimental results by applying SLWS regularization to various Mamba models. This includes evaluations under:  Supervised classification training,  Self-supervised MAE , and  Masked Feature Distillation (MFD)-based comparisons. As the reviewer mentioned, SLWS consistently *achieves measurable improvements* across these frameworks. These results demonstrate that the proposed regularization method intrinsically enhances the training of Vision Mamba models while remaining compatible with diverse training paradigms. Notably, SLWS even elevates the accuracy of the previously collapse-prone Vim-L model to a 84.5% on ImageNet, demonstrating its effectiveness.
>
> > **Q4 Table 4: Clarify the units, indicate if less/more is better**
>
> **A4:** Thanks for your suggestion and we have added such information to the revised version.
>
>
>
> ------
>
> Thank you for all your comments, which we believe have strengthened our work; we hope our responses have addressed your remaining concerns.
>
> [1] Deep networks with stochastic depth, ECCV'16.

---

### Official Review · Reviewer_mf21 · 2025-03-19

**Overall Recommendation:** 4

**Summary:**

This paper proposes a stochastic layer-wise shuffle regularization (SLWS) method for efficient vision mamba training. As a plug-and-play method, SLWS mitigates the overfitting problem with introducing minimal overhead. The achieved results are impressive and downstream tasks also verified the effectiveness. Overall, the idea is novel and interesting and the method only has minor weaknesses.

**Claims And Evidence:**

YES

**Essential References Not Discussed:**

NO

**Experimental Designs Or Analyses:**

YES

**Methods And Evaluation Criteria:**

YES

**Other Comments Or Suggestions:**

See above

**Other Strengths And Weaknesses:**

1. I think some vision mamba models have not the training issue, such as VMamba [NeurIPS'24];
2. Why the method focuses on the plain mamba models?
3. Why training for 220 epochs for some models, that is very strange. Did the author meet the overfitting problem?
4. Are there any other tricks or modules or technique during the training of your models?

**Questions For Authors:**

See above

**Relation To Broader Scientific Literature:**

Vision mamba has achieved great success in visual tasks and training process indeed a problem. This paper proposes a interesting stochastic layer-wise shuffle strategy to help the training process and achieves good results without increased overhead.

**Theoretical Claims:**

YES

---

> ### Author Rebuttal · Authors · 2025-03-31
>
> > **Q1 & Q2: Hierarchical VMamba training and non-hierarchical plain mamba selection**.
>
> **A1:** VMamba employs a more complex downsampling process and adopts a hierarchical architecture, which demonstrates strong performance under Tiny, Small, and Base model sizes. However, there is currently no evidence to confirm its scalability to larger model sizes beyond the Base size.
>
> **A2:** In contrast, plain vision Mamba models feature a non-hierarchical architecture. **1)** This design is simpler, more fundamental, easier to stack, **2)** and seamlessly compatible with a wide range of existing sequence modeling frameworks like MLLMs, as well as training paradigm like MAE and masked feature distillation. **3)** Such architectures (e.g., Vim[ICML'24], Mamba-R[CVPR'25], ARM[ICLR'25], PlainMamba[ECCV'24]) have been widely adopted and benchmarked in prior research. Building upon this foundation, we propose SLWS, a plug-and-play regularization method that further enhances the scalability of these models. Our approach achieves state-of-the-art results on ImageNet among Vision Mamba variants, contributing to the exploration of efficient regularization strategies for plain Mamba architectures.
>
>
>
> > **Q3: 220 training epoch setting on some models**.
>
> **A3:** Our training protocol for 220 epochs follows the settings of Mamba-R to ensure a fair comparison. Mamba-R adopts a three-stage training strategy, which its paper states is equivalent to approximately 220 epochs of training under a 224 input resolution. Therefore, we adhered to its protocol to demonstrate that our SLWS method is compatible with the register-based framework proposed in their work. This approach validates the seamless integration of our regularization technique with existing methodologies.
>
>
>
> > **Q4: Training configuration of our models**.
>
> **A4:** Beyond SLWS, we did not employ any special additional tricks, modules, or techniques to alter the training pipeline. All training configurations strictly adhere to the essential settings previously used in vision Mamba models, ensuring the validity of comparisons. Detailed training protocols are provided in the Appendix tables.
>
>
>
> Thank you for all your comments, which we believe have strengthened our work; we hope our responses have addressed all of your remaining concerns.

---

### Decision · Program_Chairs · 2025-05-01

**Decision:**

Accept (poster)

**Comment:**

All reviewers are in favor of acceptance for this paper, in particular they note that the proposed Stochastic Layer-Wise Shuffle (SLWS) distinguishes itself through its straightforward design and computational efficiency, achieving consistent performance improvements without the need for any architectural changes. So I m in favor of accepting this paper to ICML